# Incidence of Venous Thromboembolism in Multiple Myeloma Patients across Different Regimens: Role of Procoagulant Microparticles and Cytokine Release

**DOI:** 10.3390/jcm11102720

**Published:** 2022-05-11

**Authors:** Antonio Gidaro, Roberto Manetti, Alessandro Palmerio Delitala, Mark Johns Soloski, Giorgio Lambertenghi Deliliers, Dante Castro, Davide Soldini, Roberto Castelli

**Affiliations:** 1Department of Biomedical and Clinical Sciences Luigi Sacco, Luigi Sacco Hospital, University of Milan, Via G.B. Grassi N° 74, 20157 Milan, Italy; 2Department of Medical, Surgical and Experimental Sciences, University of Sassari, Piazza Università N° 21, 07100 Sassari, Italy; rmanetti@uniss.it (R.M.); aledelitala@uniss.it (A.P.D.); dantecastro554@gmail.com (D.C.); 3Division of Rheumatology, Johns Hopkins University School of Medicine, Baltimore, MD 21224, USA; mski@jhmi.edu; 4Fondazione Mattarelli, Largo della Crocetta, 2, 20122 Milan, Italy; giorgio.lambertenghi@unimi.it; 5Department of Internal Medicine, ASST Papa Giovanni XXIII, Piazza OMS, 1, 24127 Bergamo, Italy; dsoldini@hotmail.it

**Keywords:** multiple myeloma, thrombosis, microvesicles, TGF-β, IL-17, immunomodulatory derivatives (IMiDs), dexamethasone/bortezomib, watch and wait strategy

## Abstract

Introduction: Multiple myeloma (MM) is characterized by a high prevalence of thrombotic complications. Microvesicles (MVs) are small membrane vesicles released from activated cells, and they may potentially contribute to thrombosis. Methods: We have evaluated the plasma levels of MVs and cytokines (IL-10, IL-17, and TGF-β in MM and Watch and Wait Smoldering MM (WWSMM) from patients and related them to thrombotic complications. The secondary aim was to assess the impact of ongoing therapy on MV and on cytokine levels. Result: 92 MM and 31 WWSMM were enrolled, and 14 (12%) experienced a thrombotic episode. Using univariate analysis, TGF-β and MV were significantly higher in patients with thrombotic events (*p* = 0.012; *p* = 0.008, respectively). Utilizing a Cox proportional hazard model, we confirmed this difference (TGF-β *p* = 0.003; Odds ratio 0.001, 95% CI 0–0.003 and MV *p* = 0.001; Odds ratio 0.003, 95% CI 0.001–0.005). Active treatment management displayed higher levels of MV (*p* < 0.001) and lower levels of glomerular filtration-rate (*p* < 0.001), IL-17 (*p* < 0.001) as compared to the WWSMM group. The TGF-β values of immunomodulatory derivatives patients were lower in the WWSMM (*p* < 0.001) and Dexamethasone/Bortezomib subgroup (*p* < 0.001). Conclusion: The increased levels of MVs in active regimens add insight into the mechanisms of hypercoagulation in MM. In addition, a role for cytokine-related thrombosis is also suggested.

## 1. Introduction

Cancer patients have a 4.3-fold higher incidence of thrombotic diseases due to multiple risk factors. Among hematologic malignancies, multiple myeloma (MM) is characterized by a ~10% higher risk of developing venous thromboembolism (VTE) [1]. Increased blood viscosity due to high levels of immunoglobulin, the procoagulant activity of the monoclonal protein, and inflammatory cytokines are the main factors involved in MM-related VTE [2]. Over the last decade, advances in MM therapy have led to an increase in survival, even in relapsed/refractory MM and elderly patients [3,4,5].

After the introduction of immunomodulatory derivatives (IMiDs) as a therapeutic tool in the management of MM, VTE has emerged as one of the leading complications, particularly in newly diagnosed MM patients [6]. The incidence of VTE varies across different regimens in MM [6]. IMiDs-based treatments are associated with rates of VTE reaching values up to 14 to 26%, particularly when Dexamethasone or chemotherapy are added [7,8]. Recent studies have shown that the microenvironment in MM plays a pivotal role in disease progression and relapse [9]. For example, active crosstalk between MM cells and bone marrow stromal mesenchymal cells has been recently shown [10]. Interestingly, microvesicles (MVs), small (0.1–1 µm) membrane vesicles released from activated cells, have been identified as a soluble factor participating in intercellular communications [11]. The surface of MVs can be highly procoagulant due to the presence of the procoagulant protein tissue factor (TF) and of negatively charged phospholipids, such as phosphatidylserine [11]. TF is the most important procoagulant protein expressed by cancer cells and, together with other procoagulant factors, contributes to the thrombotic phenotype of malignant disease [11]. In patients with cancer, TF is also overexpressed by normal host blood cells triggered by cancer-derived inflammatory stimulation [11].

Therefore, a subclinical activation of blood coagulation is typically present in MM patients, as demonstrated by abnormalities of circulating thrombotic biomarkers [12]. Inflammatory cytokines promote the formation of MVs from various cell types, including endothelial cells and monocytes [11]. Most prominent among those is the proinflammatory cytokine tumor necrosis factor-alpha (TNF-α), which is used in many studies as a model agent to study MV formation. For example, experimental animal models have shown the prothrombotic activity of several cytokines, including interferon gamma (IFN)-γ, interleukin (IL)-6, IL-17A, transforming grow factor β (TGF-β chemokine (CC motif ligand 2) (CCL2), IL-9 and IL-1β. On the other hand, other cytokines such as IL-10, Tumor necrosis factor α (TNF-α), and IL-8 promote thrombus resolution [13].

The first aim of this prospective study was to correlate the plasma levels of MVs with the serum levels of immunoregulatory cytokines (TNF-α, IL-10, IL-17, and TGF-β) both in basal conditions and in patients who presented thrombotic complications. A secondary aim was to assess the impact of ongoing therapy on MV and immunoregulatory cytokines.

## 2. Materials and Methods

MM patients were serially enrolled from 30 March 2018 to 30 March 2021. The study was conducted according to the guidelines of the Declaration of Helsinki and approved by the Institutional Ethics Committee of Ospedale Maggiore Policlinico di Milano (N 206 date 29 June 2013).

The inclusion criteria were: (1) International Staging System (ISS) grade I IgG MM (2) Watch and Wait (WW) Smoldering MM. The exclusion criteria were: (1) ongoing therapy with at least one between: anti-inflammatory, anti-fibrinolytic, anti-coagulant, anti-platelet; (2) baseline risk of thrombosis: history of smoking (active or previous), history of previous arterial or venous thrombosis, chronic liver disease, grade IV chronic kidney disease (Estimated Glomerular Filtration Rate (eGFR) < 30 mL/min/1.73 m^2^), chronic inflammatory diseases, or other neoplasms.

To assess the impact of ongoing therapy on MV and immunoregulatory cytokines, the whole cohort was divided into three groups: (1) Dexamethasone and Bortezomib, (2) IMiDS-based treatment, and (3) Watch and Wait (WW) Smoldering MM. The IMiDS subgroup (1) Lenalidomide alone and (2) Melphalan, Prednisone, and Thalidomide were compared to confirm the homogeneity of the cluster.

### 2.1. Sample Collection and Storage

All blood samples were collected after 12 h of fasting and using sodium citrate 3.8% as an anti-coagulant. Antecubital venous blood samples were drawn from patients affected by MM at baseline established as the first visit for WW patients and after 1 month of therapy for patients in active treatment. Tests were performed on the same day of sample collection.

### 2.2. Measurements

Microvesicles were isolated from peripheral blood. Briefly, supernatants from the cells were centrifuged at 800× *g* for 5 min and then centrifuged at 4500× *g* for 5 min to discard large debris [14]. Microvesicles were isolated after centrifugation at 20,000× *g* for 60 min at 4 °C, followed by washing and resuspension in PBS. Ultrastructural analysis of the isolated Microvesicles was conducted using flow cytometry, including the following markers, Annexin V- PAC, anti-CD141 for platelets, anti-CD142 for tissue factors, anti-CD 144 for endothelial cells, and anti-CD138 for plasma cells. TNF-α, IL-10, IL-17, and TGF-β were measured using Sandwich ELISA immunoassays (Quantikine R&D System Inc. 614 McKinley Place NE Minneapolis, MN, USA). The Glomerular Filtration Rate (eGFR) was estimated using the CKD-EPI formula.

### 2.3. Statistical Analysis

The Kolmogorov–Smirnov test was conducted to evaluate the normality of the distribution of data. The qualitative data were expressed as both a number and a percentage. Chi-square or Fisher exact tests were used in the comparison of the groups. The quantitative data were expressed as mean, standard deviation, median, and range. The Student *t*-test and Mann–Whitney test (for non-parametric data) were used for comparison between the groups. A *p*-value less than 0.05 was considered statistically significant. The Cox proportional hazard model was used to evaluate the clinical parameters that statistically correlated with the insurgence of thrombotic complications at univariate analysis. Associations between statistically significant covariates were investigated by Pearson correlation analyses. The statistical analysis of the data was conducted using Excel (Office program 2016) and SPSS (statistical package for social science-SPSS, Inc., Chicago, IL, USA, version 20).

## 3. Results

Ninety-two myeloma patients undergoing therapy and thirty-one WW Smoldering MM patients were enrolled, 49 were men (39.8%), and 74 were women (60.2%); the median age was 74 years of age (range 70–82 years, Table 1). Twenty-nine patients were treated with Dexamethasone and Bortezomib. The IMiDS group constituted 63 patients; 32 were treated with Lenalidomide, 31 with Melphalan, Prednisone, and Thalidomide. The median duration of the follow-up was 12 months (10–16). During this time, 14 patients (12%) experienced a thrombotic episode and required hospitalization. The median time before developing a thrombotic event was 11.5 months (6.75–12). Venous thromboses included: four pulmonary embolisms, two splanchnic thromboses, and nine deep venous thromboses (VTE) of the lower limbs. Arterial thromboses included: three strokes/transient ischemic attacks, one myocardial infarction, and one retinal artery occlusion (six patients experienced more than one event).

When all of the patients were compared using univariate analysis, the levels of TGF-β and MV were significantly higher in patients with thrombotic events (*p* = 0.012; *p* = 0.008, respectively, Figure 1). Thirty-two patients (26%) had higher values of TGF-β than the normal range (344–2382 pg/mL [15]), and five of them developed thrombotic events. Eighty-four (68.3%) had higher values of MV than the normal range (1000 MVs/mL utilizing the flow cytometry technique [14]), and twelve of them developed thrombotic events.

Utilizing a Cox proportional hazard model, we confirmed this difference (Table 2: TGF-β; *p* = 0.003; odds ratio 0.001, 95% CI 0–0.003, and MV (*p* = 0.001; Odds ratio 0.003, 95% CI 0.001–0.005).

Interestingly, when we compared the levels of MV and TGF-β, we found a significant indirect association between the two in the whole cohort (Figure 2A, r = −0.496, *p* < 0.001), which was confirmed in the subgroups of patients without thrombosis (Figure 2B, r = −0.57, *p* < 0.001). Regarding patients with thrombosis, there was a trend toward a direct association, but it did not reach statistical significance (Figure 2C, r = 0.102, *p* = 0.729).

Importantly, when compared to the WW Smoldering MM, the group of patients undergoing active treatment had a significantly higher number of thrombotic events (Fisher’s test 14/92 vs. 0/31; *p* = 0.02, Table 2). Therefore, in our study, all of the thrombotic episodes were found to be on active treatment.

In addition, groups undergoing active treatment management (both the Dexamethasone/Bortezomib and IMiDs group) displayed lower levels of IL-17 (*p* < 0.001 Figure 3A), glomerular filtration-rate (*p* < 0.001 Figure 3B), and higher levels of MV (*p* < 0.001, Figure 3C) compared to the WW Smoldering MM group (Table 1). The TGF-β values of IMiDs patients were lower than the WW patients (*p* < 0.001), and the Dexamethasone and Bortezomib subgroup (*p* < 0.001 Figure 3D).

Intragroup IMiDs homogeneity was confirmed with no statistical difference in all the variables tested other than the older age in the MPT group versus Lenalidomide, respectively, 82 (77–82) vs. 74 (72–80) *p* = 0.005.

## 4. Discussion

To the best of our knowledge, this is the first report correlating circulating MVs and cytokines to thrombotic complications in MM patients treated with a specific therapy.

Two studies proposed that hypercoagulability in MM arises from thrombin generation due to clotting activation via TF and phospholipids activity [16,17]. Our result is in agreement with Auwerda et al., who detailed, that in a MM cohort under chemotherapy, elevated MV-TF activity in patients who developed VTE, in contrast to patients who do not develop VTE [17]. This elevated MV-TF activity may be associated with increased MVs, especially during anti-myeloma treatment. Importantly, our study confirms that Dexamethasone and IMiDs carry an elevated risk of VTE by increasing the levels of MVs and the related TF activation of clotting. This result needs to be validated using other larger patient cohorts.

It is known that there is a strong connection between inflammation, cancerogenesis, coagulation, and the immune system [18]. Therefore, serum levels of several key cytokines were measured in our study.

Our finding of a correlation of TGF-β levels in the development of thrombosis confirms previous studies that show an increase in patients with VTE [13]. The inflammatory effect of TGF-β and its role in fibrogenesis underlies its contribution to endothelial dysfunction and increased fibrosis [13].

TGF-β is part of a family of structurally related proteins that consists of activins/inhibins and bone morphogenic proteins [18]. Members of the TGF-β family control numerous cellular functions, including proliferation, apoptosis, differentiation, epithelial-mesenchymal transition, and migration. The first identified member, TGF-β, is implicated in several human diseases, such as vascular diseases, autoimmune disorders, and carcinogenesis. The activation of the TGF-β receptor by its ligands induces the phosphorylation of serine/threonine residues and triggers the phosphorylation of intracellular effectors (SMADs). Upon activation, SMAD proteins translocate to the nucleus and induce transcription of their target genes, regulating several cellular functions. TGF-β dysregulation has been implicated in carcinogenesis and cancer progression. In the early stages of cancer, TGF-β exhibits tumor-suppressive effects by inhibiting cell cycle progression and promoting apoptosis. However, in the late stages, TGF-β exerts tumor-promoting effects, increasing tumor invasiveness and metastasis. Furthermore, the TGF-β signaling pathway communicates with other mediators in a synergistic or antagonistic manner and regulates cellular functions. Elevated TGF-β activity has been associated with poor clinical outcomes or the advanced stages of cancer disease. In the setting of MM, TGF-β is linked with bone-related disease, and TGF-β acts as a potent immunosuppressive cytokine thought to exert effects on both cell differentiation and cell proliferation [15].

In our study, we observed a clear relationship between thrombotic events and active treatment, increased Microvesicles, and TGF-β. A study by Yamaguchi et al. [18] showed that the TGFβ1/SMAD/Plasminogen activator inhibitor-1 signaling pathway promotes the release of Tissue Factor-Bearing Microvesicles.

In our analysis of disease status and treatment regimens in MM patients, the group of MM patients in active treatment was found to have a higher number of thrombotic events and higher levels of MVs (*p* < 0.001) compared with the WW group. This confirms previous papers that have reported a higher incidence of thrombotic events in actively treated MM patients [7,8]. Chemotherapy, as well as other cellular stressors, such as heat shock, hypoxia, hypothermia, or oxidative stress, have also been reported to increase MV secretion [19].

In accordance with published studies [20,21,22], in the current study, patients in the WW myeloma group had increased levels of IL-17 compared to the active treatment cohort; this could be related to the increased amounts of IL-6 in the bone marrow of myeloma patients with active disease that promotes the production of T helper 17 cells from CD4 naive cells. In treated patients, there was a significant reduction in IL-17; the reduction in IL-17 probably reflects the disease response. Nevertheless, the relationship between T helper 17 cells and T regulatory cells in MM requires further investigation [19].

The TGF-β values were lower in the IMiDs group compared to both the Dexamethasone/Bortezomib and WW group. IMiDs agents, Thalidomide and Lenalidomide, are immunomodulatory drugs. These treatments exhibit many important therapeutic properties, such as anti-angiogenetic, antiproliferative, and pro-erythropoietic properties, whose mechanisms remain to be further clarified [23]. In addition, these drugs have shown effects on NK cytotoxic cells and cytokine production. In the literature, studies on the effect of IMiDS on TGF-β are not in agreement. In an in vitro study, Galustian et al. found that neither of these drugs (i.e., Lenalidomide and Pomalidomide) had an effect on secreted TGF-β [24]. In contrast, in agreement with our results, Hadjiaggelidou et al. demonstrated, in a small population of MM patients, a reduction in TGF-β levels in the Lenalidomide plus Dexamethasone treatment group versus those patients treated with Dexamethasone plus Bortezomib [20]. Our results confirm and extend their studies.

Treatment with high doses of Dexamethasone and Bortezomib deserves special discussion; in this group, we observed high levels of TGF-β and MVs with an increased prevalence of thrombosis. This observation, combined with the results from our IMDIS group, suggests the hypothesis of two different thrombotic pathways, one from MV-mediated TF thrombin generation and one due to the TGF-β activity.

The addition of steroids, particularly at higher doses, is associated with a significant elevation in thrombosis risk. Rajkumar [25] reported a comparison between Lenalidomide with high- or low-dose Dexamethasone. In this study, a high-dose was referred to as 40 mg of Dexamethasone on days 1–4, 9–12, and 17–20 of a 28-day cycle, versus a low dose, where 40 mg of Dexamethasone was administered once weekly. The total dose of Dexamethasone received in the ‘high-dose’ group was 480 mg/month, in line with the International Myeloma Working Group’s (IMWG’s) later definition of ‘high-dose’ glucocorticoids [26]. In the initial part of the study, VTE prophylaxis was recommended but not mandated. Of the first 266 enrolled patients, 18.2% developed VTE in the high-dose group and 3.7% in the low-dose group, after which thromboprophylaxis became mandatory [25]. At 1 year from study initiation, the VTE rate in the high-dose group was over double that of the low-dose group (26% vs. 12%), providing substantial supportive evidence for the thrombogenic potential of high dose Dexamethasone [27].

Lastly, we note that the present study has several limitations, including the number of patients enrolled to investigate the risk of thrombosis and the low numbers of inflammatory cytokines investigated.

In addition, blood samples were drawn at baseline (at study entry for patients on WW Smoldering MM vs. 1 month of treatment for patients on active therapy), but then no subsequent or serial blood analysis was conducted. However, most thromboembolism episodes occurred many months after study enrollment, so we cannot rule out a change in the variable tested during the follow-up period.

## 5. Conclusions

To conclude, this is the first prospective study correlating MVs levels with different inflammatory cytokines in MM. Overall, we found that MVs increase in treated MM patients across different regimens. The increased levels of MVs in these regimens add insight into mechanisms of hypercoagulation in MM. In addition, the role of cytokine-related thrombosis is also suggested and remains to be further investigated.

## Figures and Tables

**Figure 1 jcm-11-02720-f001:**
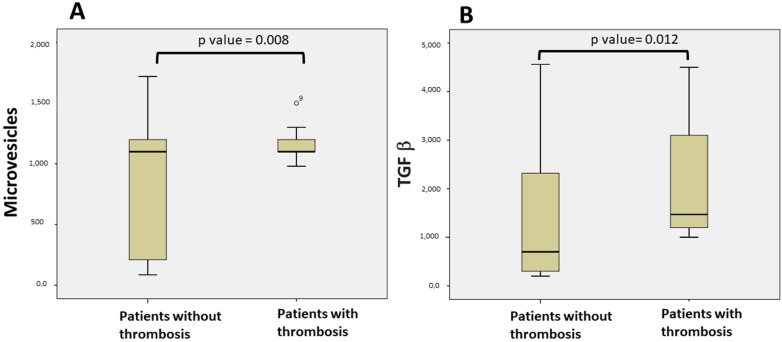
Comparison of patients with thrombosis and without thrombosis: (**A**) Microvesicle levels in patients with thrombosis and patients without thrombosis; (**B**) TGF-β levels in patients with thrombosis and patients without thrombosis.

**Figure 2 jcm-11-02720-f002:**
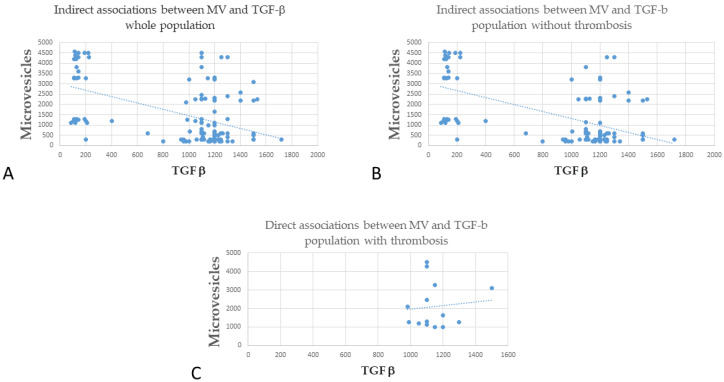
Correlation between MV and TGF-β (**A**) Indirect association between MV and TGF-β whole population; (**B**) Indirect association between MV and TGF-β population without thrombosis). (**C**) Direct associations between MV and TGF-β in the population with thrombosis.

**Figure 3 jcm-11-02720-f003:**
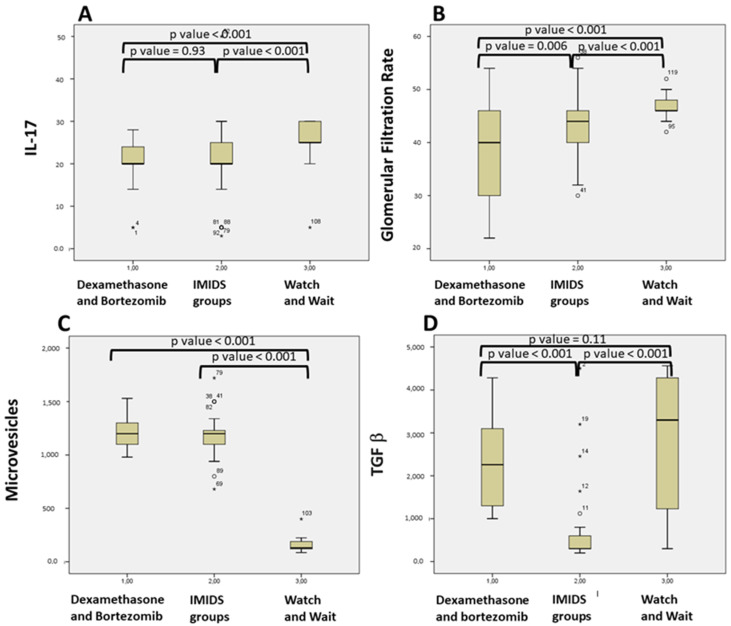
Comparison of patients based on different treatment: (**A**) IL-17 levels in Dexamethasone and Bortezomib; IMiDs based treatment; Watch and Wait Smoldering MM patients; (**B**) Glomerular Filtration Rate levels in Dexamethasone and Bortezomib; IMiDs based treatment; Watch and Wait Smoldering MM patients (**C**) Microvesicles in Dexamethasone and Bortezomib; IMiDs based treatment; Watch and Wait Smoldering MM patients (**D**) TGF-β in Dexamethasone and Bortezomib; IMiDs based treatment; Watch and Wait Smoldering MM patients.

**Table 1 jcm-11-02720-t001:** Laboratory exams and inflammatory parameters in: A. whole cohort B. Dexamethasone and Bortezomib C. IMiDs D. Watch and Wait Smoldering MM. Data are reported as “median (IQR)”.

	A. Whole Population	B. Dexamethasone and Bortezomib	C. IMiDs	D. Watch and Waits Smoldering MM	*p* Value B vs. C	*p* Value B vs. D	*p* Value C vs. D
Age [years]	74 (70–82)	68 (67–73.5)	80 (74–82)	72 (70–80)	**<0.001**	**0.025**	**0.002**
Number of patients with thrombosis/number of patients	14/123	10/29	4/63	0/31	**0.002**	**0.017**	0.3
● Melphalan, Prednisone and Thalidomide (MPT)			4/32				
● Lenalidomide			0/31				
Median follow-up [months]	12 (10–16)	12 (8–13)	12 (8–13)	20 (12–32)			
Microvescicles [N/mL]	1100 (400–1200)	1200 (1100–1300)	1200 (1100–1230)	130 (120–190)	0.83	**<0.001**	**<0.001**
Hemoglobin [g/dL]	10.2 (9.6–11.3)	10.4 (9.4–11.4)	10.4 (9.6–11.4)	10 (9.7–10.7)	0.65	0.77	0.4
Glomerular Filtration Rate (eGFR) [mL/min/1.73 m^2^]	46 (40–48)	40 (30–46)	44 (40–46)	46 (46–48)	**0.006**	**<0.001**	**<0.001**
Number of patients with Chronic kidney disease (CKD) Stage 3a: 45 to 59 [mL/min/1.73 m^2^] eGFR	68	9	30	29	0.17	**<0.001**	**<0.001**
Number of patients with CKD Stage 3b: 30 to 44 [mL/min/1.73 m^2^] eGFR	49	20	33	2	0.17	**<0.001**	**<0.001**
Platelets [×10^9^/L]	187,000 (142,000–210,000)	203,000 (126,000–230,000)	178,000 (142,000–203,000)	164,000 (123,000–242,500)	0.27	0.9	0.9
Activated Partial Thromboplastin Time (aPTT)	1.02 (0.96–1.08)	1 (0.96–1.06)	1.03 (0.97–1.06)	1.01 (0.97–1.08)	0.91	0.95	0.97
International Normalized Ratio (INR)	1 (0.95–1.05	1.01 (0.95–1.03)	0.99 (0.96–1.03)	1.01 (0.95–1.05)	0.97	0.91	0.93
TNF-α [pg/mL]	1 (1–2)	2 (1–2)	2 (1–2)	1 (1–2)	0.14	**0.029**	0.14
IL-17 [pg/mL]	20 (20–25)	20 (18–24)	20 (20–25)	25 (25–30)	0.93	**<0.001**	**<0.001**
TGF-β pg/mL	1100 (300–2456)	2259 (1290–3150)	300 (300–600)	3300 (1200–4280)	**<0.001**	0.11	**<0.001**
Monoclonal protein level [g/dL]	3 (2.6–3.5)	2.6 (2.4–3.5)	3 (3–3.5)	3 (3–3.5)	**0.037**	0.1	0.97
IL-10 [pg/mL]	12 (6–35)	6 (2–10)	25 (7–60)	12 (12–12)	**<0.001**	**<0.001**	0.266

**Table 2 jcm-11-02720-t002:** MVs and serum levels of immunoregulatory cytokines (TNF-α, IL-10, IL-17, and TGF-β) in: A. Whole cohort B. Patients with thrombosis C. Patients without thrombosis. Data are reported as “median (IQR)”.

	A. Whole Population	B. Patients with Thrombosis	C. Patients without Thrombosis	*p* Value Univariate	*p* Value Multivariate Cox Proportional Hazard Model	Odds Ratio Multivariate
Age [years]	74 (70–82)	71.5 (67.75–77.25)	74 (70–82)	0.713		
Number of patients	123	14	109			
Microvescicles [N/mL]	1100 (400–1200)	1100 (1087–1200)	1100 (200–1200)	**0.008**	**0.001**	**0.003 (0.001–0.005)**
TNF α [pg/mL]	1 (1–2)	1.5 (1–2)	1 (1–2)	0.228		
IL-17 [pg/mL]	20 (20–25)	20 (14–21.25)	20 (20–25)	0.085		
TGF-β ng/mL	1100 (300–2456)	1470 (1180–3145)	700 (300–2360)	**0.012**	**0.003**	**0.001 (0–0.003)**
IL-10 [pg/mL]	12 (6–35)	7.5 (4.25–13.5)	12 (6.5–36)	0.181		

## Data Availability

The study data will be made available upon request to the corresponding author.

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
