# Peer review of "Incidence of Venous Thromboembolism in Multiple Myeloma Patients across Different Regimens: Role of Procoagulant Microparticles and Cytokine Release"

_jcm, 2022, doi:10.3390/jcm11102720_

Round 1
Reviewer 1 Report
Dear Authors,
it was a pleasure for me to read and review such an interesting and well-conducted paper. I appreciate your study, however I have some suggestions.
Firstly, I would like to suggest adding more information about patients' myeloma status like myeloma type and ISS staging.
Secondly, in my opinion the data about patients' previous medical history of thrombotic events could be described to assess baseline risk of thrombosis.
Additionally, I would like to enclose the blood clotting parameters like APTT or INR to analysis.
Could you also describe the mathematical formula that was used to calculate the GFR (MDRD, CKD-EPI etc) and add the data about the prevalence of renal failure to the table, please.
Please, check the punctuation (dots, commas, space) in a whole text.
Finally, I will recommend the Editors to accept your paper after minor revision.
Author Response
Answers to Reviewer 1
Reviewer 1: Dear Authors,
it was a pleasure for me to read and review such an interesting and well-conducted paper. I appreciate your study; however, I have some suggestions.
Response to Reviewer 1 comment: We thank the reviewer for his/her positive comment.
Reviewer 1: Firstly, I would like to suggest adding more information about patients' myeloma status like myeloma type and ISS staging.
Response to Reviewer 1 comment: We thank the reviewer for his/her suggestion. Now we specified in the inclusion criteria that all Myeloma patients were International Staging System (ISS) grade I IgG MM or Watch and Wait (WW) Smoldering MM.
Reviewer 1: Secondly, in my opinion the data about patients' previous medical history of thrombotic events could be described to assess baseline risk of thrombosis.
Response to Reviewer 1 comment: We thank the reviewer for his/her suggestion. Now we specified in the exclusion criteria that all Myeloma patients were without (1) ongoing therapy with at least one between: anti-inflammatory, anti-fibrinolytic, anti-coagulant, anti-platelet; (2) baseline risk of thrombosis: history of smoking (active or previous), history of previous arterial or venous thrombosis, chronic liver disease, grade IV chronic kidney disease (Es-timated Glomerular Filtration Rate (eGFR)<30 mL/min/1.73 m2), chronic inflammatory diseases, or other neoplasms.
Reviewer 1: Additionally, I would like to enclose the blood clotting parameters like APTT or INR to analysis.
Response to Reviewer 1 comment: We thank the reviewer for his/her suggestion. Now we add APTT and INR to analysis
Reviewer 1: Could you also describe the mathematical formula that was used to calculate the GFR (MDRD, CKD-EPI etc) and add the data about the prevalence of renal failure to the table, please.
Response to Reviewer 1 comment: We thank the reviewer for his/her suggestion. We specified the use of CKD-EPI in the methods section. The prevalence of renal failure is added to the table.
Reviewer 1: Please, check the punctuation (dots, commas, space) in a whole text.
Response to Reviewer 1 comment: We thank the reviewer for his/her correction. Now we perform a thorough English revision.
Reviewer 1: Finally, I will recommend the Editors to accept your paper after minor revision.
Response to Reviewer 1 comment: We thank the reviewer for his/her positive comment.
Answers to Reviewer 2
Reviewer 2: The authors presented a prospective study aiming to assess the levels of microvesicles and different interleukins and their association with thromboembolic complications in patients with multiple myeloma treated with different combinations of therapy. As the incidence of such TE events is very high in MM patients I believe the topic is important and needs exploration. However during the review process I was able to identify many areas needing improvement. Hereby I list the issues in order of their finding in the article:
Response to Reviewer 2 comment: We thank the reviewer for his/her positive comment.
Reviewer 2: Please try to make the title shorter and correct the spelling mistake (Micropartcels->Microparticles).
Response to Reviewer 2 comment: We thank the reviewer for his/her suggestion. Now the Title change as reviewer recommend.
Reviewer 2: Patients with multiple myeloma need treatment from the diagnosis, as it requires a CRAB organ involvement. Earlier it is called MGUS or Smoldering multiple myeloma. How did you distinguish the Watch and Wait group?
Response to Reviewer 2 comment: We thank the reviewer for his/her suggestion. Now we specified in the inclusion criteria that all Myeloma patients were International Staging System (ISS) grade I IgG MM or Watch and Wait (WW) Smoldering MM.
Reviewer 2: Please use assess instead of weigh throughout the article, as it is not correct.
Response to Reviewer 2 comment: We thank the reviewer for his/her correction. Now we use assess instead of weigh throughout the article.
Reviewer 2: Please use . instead of , in numerical characters. Please delete double-spaces.
Response to Reviewer 2 comment: We thank the reviewer for his/her correction. Now we use . instead of , in numerical character, delete double-spaces have been deleted
Reviewer 2: Please perform a thorough English revision, best with the use of an English native speaker, as I was able to identify more than 50 spelling and grammar errors in the article.
Response to Reviewer 2 comment: We thank the reviewer for his/her correction. Now we perform a thorough English revision.
Reviewer 2: The use of whole population is unfortunate, as it sounds as if you checked healthy individuals as a control group. Please change it.
Response to Reviewer 2 comment: We thank the reviewer for his/her correction. Now we use whole cohort instead of whole population
Reviewer 2: Please use the same size of font in figures X-axis names.
Response to Reviewer 2 comment: We thank the reviewer for his/her correction. Now we use the same size of font in figures X-axis names
Reviewer 2: Please do not use ALL CAPITAL LETTERS in the table names.
Response to Reviewer 2 comment: We thank the reviewer for his/her correction. Now we use miniscule in the table names
Reviewer 2: The MPT treatment is known to have much higher thromboembolic potential, than Lenalidomide alone, hence creating the group treated with IMiDs can be very heterogeneous. Maybe dividing the group for Thalidomide-based regimen and Lenalidomide-based regimen groups and their comparison could show some interesting results.
Response to Reviewer 2 comment: We thank the reviewer for his/her suggestion. Now we add this sentence “Intragroup IMIDS homogeneity was confirmed by no statistical difference in all the variable tested other than the older age in the MPT group versus Lenalidomide, respectively 82 (77-82) vs 74 (72 – 80) p=0.005”
Reviewer 2: The sentence: Furthermore, Dexamethasone and Bortezomib treatment have a higher risk of thrombosis compared both with IMIDS groups (p=0,001; Odds ratio 7,8 [2,2 to 27,7]) and WW 158 (p=0,017; Odds ratio 33,9 [1,9 to 612], TABLE 1). is generally true, however when written as a result should be more conservative, as you do not have enough data to obtain such general conclusion.
Response to Reviewer 2 comment: We thank the reviewer for his/her correction. Now the sentence was deleted.
Reviewer 2: Monoclonal gammopathy should be changed to Monoclonal protein level in Table 1.
Response to Reviewer 2 comment: We thank the reviewer for his/her correction. Now we use Monoclonal protein level instead of Monoclonal gammopathy
Reviewer 2: In Table 2 the "Whole population" column does not have the number of patients. Moreover range for Microvescicles in "Patients without thrombosis" is 200-1200, whereas the range for "Whole population" is (400-1200). Please check the statistics for mistakes.
Response to Reviewer 2 comment: We thank the reviewer for his/her correction. Now we add the number of patients. Statistic was check for mistakes
Answers to Reviewer 3
Reviewer 3: Multiple myeloma is known to be associated with an increased risk of thromboembolism; some of this risk is due to the underlying disease and some due to MM-specific therapies. There is growing recognition of the role of microvesicles in thromboembolism and this is an area where further work needs to be done. This article summarizes a study where plasma levels of cytokines and microvesicles were obtained in patients with MM. The study includes MM patients on therapy and MM patients on observation as a comparator group.
Comments:
- The methods section seems to indicate that blood samples were drawn at baseline (at study entry for watch/wait vs 1 month into treatment for active therapy patients), but then no subsequent or serial blood samples were drawn for either group. Yet most episodes of thromboembolism occurred many months after these blood samples were drawn. Therefore, there is an implicit assumption that the blood levels of cytokines and microvesicles obtained early on have remained stable, but we do not know if this is the case or not because they apparently were never rechecked. For example, what if some patients had progressive myeloma or a treatment response or came off of anti-myeloma therapy? This is a significant study design limitation and needs to be commented on by the authors.
Response to Reviewer 3 comment: We thank the reviewer for his/her suggestion. Now this limitation has been added in the discussion section. This sentence was added: “In addition, blood samples were drawn at baseline (at study entry for patients on watch/wait vs 1 month of treatment for patients on active therapy), but then no subsequent or serial blood analysis was conducted. Yet most thromboembolism episodes occurred many months of study enrollment, so we cannot rule out a change in the variable tested during the follow-up period.”
Reviewer 3: There are other known risk factors for venous and arterial thromboembolism aside from a diagnosis of multiple myeloma and certain anti-myeloma treatments. There is no comment in the manuscript on which of these risk factors the patients in this study may or may not have. Furthermore, there is no comment on whether patients were on medicines to reduce the risk of thromboembolism such as aspirin. Scientifically, the presence of a correlation between TGF-B and microvesicle levels and increased risk of thromboembolism may be misleading if the baseline risk of thromboembolism is unknown. The authors need to comment on their knowledge or lack thereof of these potential confounders in the patient population. If this data is available, then the conclusions could be potentially strengthened and made more meaningful.
Response to Reviewer 3 comment: We thank the reviewer for his/her suggestion. Now we specified in the exclusion criteria that all Myeloma patients were without (1) ongoing therapy with at least one between: anti-inflammatory, anti-fibrinolytic, anti-coagulant, anti-platelet; (2) baseline risk of thrombosis: history of smoking (active or previous), history of previous arterial or venous thrombosis, chronic liver disease, grade IV chronic kidney disease (Estimated Glomerular Filtration Rate (eGFR)<30 mL/min/1.73 m2), chronic inflammatory diseases, or neoplasms.
Reviewer 3: As a corollary to point 2, the finding that most episodes of thromboembolism were in the bortezomib / dexamethasone treatment group is presumably unexpected, as most of the literature thus far has indicated that bortezomib is not thought to be associated with an increased risk compared to other myeloma therapies. Was this finding due to the presence of other risk factors in the bortezomib patient population? Was this finding due to the presence of other risk factors in the bortezomib patient population? Was it the dexamethasone usage that drove this finding? Did patients receiving lenalidomide receive dexamethasone?
Response to Reviewer 3 comment: We thank the reviewer for his/her suggestion. Patients receiving lenalidomide didn’t receive dexamethasone. We hypothesize that dexamethasone usage drove this finding, this sentence was added: “The addition of steroids, particularly at higher doses, is associated with a significant elevation in thrombosis risk. Rajkumar [26] reported a comparison between lenalidomide with high- or low-dose dexamethasone. In this study, high-dose was referred to as 40 mg dexamethasone on days 1–4, 9–12 and 17–20 of a 28-day cycle, versus low-dose where 40 mg of dexamethasone was administered once weekly. The total dose of dexamethasone received in the ‘high-dose’ group was 480 mg/month, in line with the international Myeloma Working Group’s (IMWG’s) later definition of ‘high-dose’ glucocorticoids [27]. In the initial part of the study, VTE prophylaxis was recommended but not mandated. Of the first 266 enrolled patients, 18.2% developed VTE in the high-dose group and 3.7% in the low-dose group, after which thromboprophylaxis became mandatory [26]. At 1 year from study initiation, the VTE rate in the high-dose group was over double that of the low-dose group (26% vs. 12%), providing substantial supportive evidence for the thrombogenic potential of high dose dexamethasone [28].”

Reviewer 2 Report
The authors presented a prospective study aiming to assess the levels of microvesicles and different interleukins and their association with thromboembolic complications in patients with multiple myeloma treated with different combinations of therapy. As the incidence of such TE events is very high in MM patients I believe the topic is important and needs exploration. However during the review process I was able to identify many areas needing improvement. Hereby I list the issues in order of their finding in the article:
- Please try to make the title shorter and correct the spelling mistake (Micropartcels->Microparticles).
- Patients with multiple myeloma need treatment from the diagnosis, as it requires a CRAB organ involvement. Earlier it is called MGUS or smaldering multiple myeloma. How did you distinguish the Watch and Wait group?
- Please use assess instead of weigh throughout the article, as it is not correct.
- Please use . instead of , in numerical characters. Please delete double-spaces.
- Please perform a thorough English revision, best with the use of an English native speaker, as I was able to identify more than 50 spelling and grammar errors in the article.
- The use of whole population is unfortunate, as it sounds as if you checked healthy individuals as a control group. Please change it.
- Please use the same size of font in figures X-axis names.
- Please do not use ALL CAPITAL LETTERS in the table names.
- The MPT treatment is known to have much higher thromboembolic potential, than Lenalidomide alone, hence creating the group treated with IMiDs can be very heterogenous. Maybe dividing the group for Thalidomide-based regimen and Lenalidomide-based regimen groups and their comparison could show some interesting results.
- The sentence: Furthermore, Dexamethasone and Bortezomib treatment have a higher risk of throm-157 bosis compared both with IMIDS groups (p=0,001; Odds ratio 7,8 [2,2 to 27,7]) and WW 158 (p=0,017; Odds ratio 33,9 [1,9 to 612], TABLE 1). is generally true, however when written as a result should be more conservative, as you do not have enough data to obtain such general conclusion.
- Monoclonal gammopathy should be changed to Monoclonal protein level in Table 1.
- In Table 2 the "Whole population" column does not have the number of patients. Moreover range for Microvescicles in "Patients without thromosis" is 200-1200, whereas the range for "Whole population" is (400-1200). Please check the statistics for mistakes.
Author Response

(The authors gave the same response as above.)

Reviewer 3 Report
Multiple myeloma is known to be associated with an increased risk of thromboembolism; some of this risk is due to the underlying disease and some due to MM-specific therapies. There is growing recognition of the role of microvesicles in thromboembolism and this is an area where further work needs to be done. This article summarizes a study where plasma levels of cytokines and microvesicles were obtained in patients with MM. The study includes MM patients on therapy and MM patients on observation as a comparator group.
Comments:
- The methods section seems to indicate that blood samples were drawn at baseline (at study entry for watch/wait vs 1 month into treatment for active therapy patients), but then no subsequent or serial blood samples were drawn for either group. Yet most episodes of thromboembolism occurred many months after these blood samples were drawn. Therefore, there is an implicit assumption that the blood levels of cytokines and microvesicles obtained early on have remained stable, but we do not know if this is the case or not because they apparently were never rechecked. For example, what if some patients had progressive myeloma or a treatment response or came off of anti-myeloma therapy? This is a significant study design limitation and needs to be commented on by the authors.
- There are other known risk factors for venous and arterial thromboembolism aside from a diagnosis of multiple myeloma and certain anti-myeloma treatments. There is no comment in the manuscript on which of these risk factors the patients in this study may or may not have. Furthermore, there is no comment on whether patients were on medicines to reduce the risk of thromboembolism such as aspirin. Scientifically, the presence of a correlation between TGF-B and microvesicle levels and increased risk of thromboembolism may be misleading if the baseline risk of thromboembolism is unknown. The authors need to comment on their knowledge or lack thereof of these potential confounders in the patient population. If this data is available, then the conclusions could be potentially strengthened and made more meaningful.
- As a corollary to point 2, the finding that most episodes of thromboembolism were in the bortezomib / dexamethasone treatment group is presumably unexpected, as most of the literature thus far has indicated that bortezomib is not thought to be associated with an increased risk compared to other myeloma therapies. Was this finding due to the presence of other risk factors in the bortezomib patient population? Was this finding due to the presence of other risk factors in the bortezomib patient population? Was it the dexamethasone usage that drove this finding? Did patients receiving lenalidomide receive dexamethasone?
Author Response

(The authors gave the same response as above.)

Round 2
Reviewer 3 Report
Authors have addressed the points raised. I have no further comments. Outside of minor spelling and grammatical review, I believe manuscript is acceptable.